# DATA AUGMENTATION INSTEAD OF EXPLICIT REGULARIZATION

## ABSTRACT

Modern deep artificial neural networks have achieved impressive results through models with orders of magnitude more parameters than training examples which control overfitting with the help of regularization. Regularization can be implicit, as is the case of stochastic gradient descent and parameter sharing in convolutional layers, or explicit. Explicit regularization techniques, most common forms are weight decay and dropout, have proven successful in terms of improved generalization, but they introduce sensitive hyper-parameters and, incongruously, often require deeper and wider architectures to compensate for the reduced capacity. In contrast, data augmentation techniques exploit domain knowledge to increase the number of training examples and improve generalization without reducing the representational capacity and without introducing model-dependent parameters, since it is applied on the training data. In this paper we systematically contrast data augmentation and explicit regularization on three popular architectures and three image object classification data sets. Our results demonstrate that data augmentation alone can achieve the same performance or higher as regularized models and exhibits much higher adaptability to changes in the architecture and the amount of training data.

## 1 INTRODUCTION

One of the central issues in machine learning research and application is finding ways of improving generalization. Regularization, loosely defined as any modification applied to a learning algorithm that helps prevent overfitting, plays therefore a key role in machine learning (Girosi et al., 1995; Müller, 2012). In the case of deep learning, where neural networks tend to have several orders of magnitude more parameters than training examples, statistical learning theory (Vapnik & Chervonenkis, 1971) indicates that regularization becomes even more crucial. Accordingly, a myriad of techniques have been proposed as regularizers: weight decay (Hanson & Pratt, 1989) and other $L^p$ penalties; dropout (Srivastava et al., 2014) and stochastic depth (Huang et al., 2016), to name a few examples. Moreover, whereas in simpler machine learning algorithms the regularizers can be easily identified as explicit terms in the objective function, in modern deep neural networks the sources of regularization are not only explicit, but implicit (Neyshabur et al., 2014). In this regard, many techniques have been studied for their regularization effect, despite not being explicitly intended as such. That is the case of unsupervised pre-training (Erhan et al., 2010), multi-task learning (Caruana, 1998), convolutional layers (LeCun et al., 1990), batch normalization (Ioffe & Szegedy, 2015) or adversarial training (Szegedy et al., 2013). In sum, there are multiple elements in deep learning that contribute to reduce overfitting and thus improve generalization.

Driven by the success of such techniques and the efficient use of GPUs, considerable research effort has been devoted to finding ways of training deeper and wider networks with larger capacity (Simonyan & Zisserman, 2014; He et al., 2016; Zagoruyko & Komodakis, 2016). Ironically, the increased representational capacity is eventually reduced in practice by the use of explicit regularization, most commonly weight decay and dropout. It is known, for instance, that the gain in generalization provided by dropout comes at the cost of using larger models and training for longer (Goodfellow et al., 2016). Hence, it seems that with these standard regularization methods deep networks are wasting capacity (Dauphin & Bengio, 2013).

Unlike explicit regularization, data augmentation improves generalization without reducing the capacity of the model. Data augmentation, that is synthetically expanding a data set by apply-

ing transformations on the available examples, has been long used in machine learning (Simard et al., 1992) and identified as a critical component of many recent successful models, like AlexNet (Krizhevsky et al., 2012), All-CNN (Springenberg et al., 2014) or ResNet (He et al., 2016), among others. Although it is most popular in computer vision, data augmentation has also proven effective in speech recognition (Jaitly & Hinton, 2013), music source separation (Uhlich et al., 2017) or text categorization (Lu et al., 2006). Today, data augmentation is an almost ubiquitous technique in deep learning, which can also be regarded as an implicit regularizer for it improves generalization.

Recently, the deep learning community has become more aware of the importance of data augmentation (Hernández-García & König, 2018b) and new techniques, such as cutout (DeVries & Taylor, 2017a) or augmentation in the feature space (DeVries & Taylor, 2017b), have been proposed. Very interestingly, a promising avenue for future research has been set by recently proposed models that automatically learn the data transformations (Hauberg et al., 2016; Lemley et al., 2017; Ratner et al., 2017; Antoniou et al., 2017). Nonetheless, another study by Perez & Wang (2017) analyzed the performance of different techniques for object recognition and concluded that one of the most successful techniques so far is still the traditional data augmentation carried out in most studies.

However, despite its popularity, the literature lacks, to our knowledge, a systematic analysis of the impact of data augmentation on convolutional neural networks compared to explicit regularization. It is a common practice to train the models with both explicit regularization, typically weight decay and dropout, and data augmentation, assuming they all complement each other. Zhang et al. (2017) included data augmentation in their analysis of generalization of deep networks, but it was questionably considered an explicit regularizer similar to weight decay and dropout. To our knowledge, the first time data augmentation and explicit regularization were systematically contrasted was the preliminary study by Hernández-García & König (2018b). The present work aims at largely extending that work both with more empirical results and a theoretical discussion.

Our specific contributions are the following:

- Propose definitions of explicit and implicit regularization that aim at solving the ambiguity in the literature (Section 2).
- A theoretical discussion based on statistical learning theory about the differences between explicit regularization and data augmentation, highlighting the advantages of the latter (Section 3).
- An empirical analysis of the performance of models trained with and without explicit regularization, and different levels of data augmentation on several benchmarks (Sections 4 and 5). Further, we study their adaptability to learning from fewer examples (Section 5.2) and to changes in the architecture (Section 5.3).
- A discussion on why encouraging data augmentation instead of explicit regularization can benefit both theory and practice in deep learning (Section 6).

## 2   EXPLICIT AND IMPLICIT REGULARIZATION

Zhang et al. (2017) raised the thought-provoking idea that "explicit regularization may improve generalization performance, but is neither necessary nor by itself sufficient for controlling generalization error." The authors came to this conclusion from the observation that turning off the explicit regularizers of a model does not prevent the model from generalizing reasonably well. This contrasts with traditional machine learning involving convex optimization, where regularization is necessary to avoid overfitting and generalize (Vapnik & Chervonenkis, 1971). Such observation led the authors to suggest the need for "rethinking generalization" in order to understand deep learning.

We argue it is not necessary to rethink generalization if we instead rethink regularization and, in particular, data augmentation. Despite their thorough analysis and relevant conclusions, Zhang et al. (2017) arguably underestimated the role of implicit regularization and considered data augmentation an explicit form of regularization much like weight decay and dropout. This illustrates that the terms explicit and implicit regularization have been used subjectively and inconsistently in the literature before. In order to avoid the ambiguity and facilitate the discussion, we propose the following definitions of explicit and implicit regularization[1]:

---

[1]See Appendix E for a short, additional discussion on the regularization taxonomy

- **Explicit regularization techniques** are those which reduce the *representational* capacity of the model they are applied on. That is, given a model class $\mathcal{H}_0$, for instance a neural network architecture, the introduction of explicit regularization will span a new hypothesis set $\mathcal{H}_1$, which is a proper subset of the original set, i.e. $\mathcal{H}_1 \subsetneq \mathcal{H}_0$.

- **Implicit regularization** is the reduction of the generalization error or overfitting provided by means other than explicit regularization techniques. Elements that provide implicit regularization do not reduce the *representational* capacity, but may affect the *effective* capacity of the model, that is the achievable set of hypotheses given the model, the optimization algorithm, hyperparameters, etc.

One of the most common explicit regularization techniques in machine learning is $L^p$-norm regularization, of which weight decay is a particular case, widely used in deep learning. Weight decay sets a penalty on the $L^2$ norm of the learnable parameters, thus constraining the representational capacity of the model. Dropout is another common example of explicit regularization, where the hypothesis set is reduced by stochastically deactivating a number of neurons during training. Similar to dropout, stochastic depth, which drops whole layers instead of neurons, is also an explicit regularization technique.

There are multiple elements in deep neural networks that implicitly regularize the models. Note, in this regard, that the above definition, contrary to explicit regularization, does not refer to *techniques*, but to a regularization *effect*, as it can be provided by elements of very different nature. For instance, stochastic gradient descent (SGD) is known to have an implicit regularization effect without constraining the representational capacity. Batch normalization does not either reduce the capacity, but it improves generalization by smoothing the optimization landscape Santurkar et al. (2018). Of quite a different nature, but still implicit, is the regularization effect provided by early stopping, which does not reduce the representational, but the effective capacity.

By analyzing the literature, we identified some previous pieces of work which, lacking a definition of explicit and implicit regularization, made a distinction apparently based on the mere intention of the practitioner. Under such notion, data augmentation has been considered in some cases an explicit regularization technique, as in Zhang et al. (2017). Here, we have provided definitions for explicit and implicit regularization based on their effect on the representational capacity and argue that data augmentation is not explicit, but implicit regularization, since it does not affect the representational capacity of the model.

## 3 THEORETICAL INSIGHTS

The generalization of a model class $\mathcal{H}$ can be analyzed through complexity measures such as the VC-dimension or, more generally, the Rademacher complexity $\mathcal{R}_n(\mathcal{H}) = \mathbb{E}_{S \sim D^n}\left[\hat{\mathcal{R}}_S(\mathcal{H})\right]$, where:

$$\hat{\mathcal{R}}_S(\mathcal{H}) = \mathbb{E}_\sigma \left[ \sup_{h \in \mathcal{H}} \left| \frac{1}{n} \sum_{i=1}^{n} \sigma_i h(x_i) \right| \right]$$

is the empirical Rademacher complexity, defined with respect to a set of data samples $S = (x_i, ..., x_n)$. Then, in the case of binary classification and the class of linear separators, the generalization error of a hypothesis, $\hat{\epsilon}_S(h)$, can be bounded using the Rademacher complexity:

$$\hat{\epsilon}_S(h) \leq \mathcal{R}_n(\mathcal{H}) + \mathcal{O}\left( \sqrt{\frac{\ln 1/\delta}{n}} \right) \tag{1}$$

with probability $1 - \delta$. Tighter bounds for some model classes, such as fully connected neural networks, can be obtained (Bartlett & Mendelson, 2002), but it is not trivial to formally analyze the influence on generalization of specific architectures or techniques.

Nonetheless, we can use these theoretical insights to discuss the differences between explicit regularization—particularly weight decay and dropout—and implicit regularization—particularly

data augmentation. A straightforward yet very relevant conclusion from the analysis of any generalization bound is the strong dependence on the number of training examples $n$. Increasing $n$ drastically improves the generalization guarantees, as reflected by the second term in RHS of Equation 1 and the dependence of the Rademacher complexity (LHS) on the sample size as well. Data augmentation exploits prior knowledge of the data domain $D$ to create new examples and its impact on generalization is related to an increment in $n$, since stochastic data augmentation can generate virtually infinite different samples. Admittedly, the augmented samples are not independent and identically distributed and thus, the effective increment of samples does not strictly correspond to the increment in $n$. This is why formally analyzing the impact of data augmentation on generalization is complex and out of the scope of this paper. Recently, some studies have taken steps in this direction by analyzing the effect of simplified data transformations on generalization from a theoretical point of view Chen et al. (2019); Rajput et al. (2019).

In contrast, explicit regularization methods aim, in general, at improving the generalization error by constraining the hypothesis class $\mathcal{H}$, which hopefully should reduce its complexity, $\mathcal{R}_n(\mathcal{H})$, and, in turn, the generalization error $\hat{\epsilon}_S(h)$. Crucially, while data augmentation exploits domain knowledge, most explicit regularization methods only naively constrain the hypothesis class. For instance, weight decay constrains the learnable models $\mathcal{H}$ by setting a penalty on the weights norm. However, Bartlett et al. (2017) have recently shown that weight decay has little impact on the generalization bounds and confidence margins.

Dropout has been extensively used and studied as a regularization method for neural networks (Wager et al., 2013), but the exact way in which dropout may improve generalization is still an open question and it has been concluded that the effects of dropout on neural networks are somewhat mysterious, complicated and its penalty highly non-convex (Helmbold & Long, 2017). Recently, Mou et al. (2018) have established new generalization bounds on the variance induced by a particular type of dropout on feedforward neural network. Nevertheless, dropout can also be analyzed as a random form of data augmentation without domain knowledge Bouthillier et al. (2015), that is data-dependent regularization. Therefore, any generalization bound derived for dropout can be regarded as a pessimistic bound for domain-specific, standard data augmentation.

A similar argument applies for weight decay, which, as first shown by Bishop (1995), is equivalent to training with noisy examples if the noise amplitude is small and the objective is the sum-of-squares error function. In sum, many forms of explicit regularization are at least approximately equivalent to adding random noise to the training examples, which is the simplest form of data augmentation[2]. Thus, it is reasonable to argue that more sophisticated data augmentation can overshadow the benefits provided by explicit regularization.

In general, we argue that the reason why explicit regularization may not be necessary is that neural networks are already implicitly regularized by many elements—stochastic gradient descent (SGD), convolutional layers, normalization and data augmentation, to name a few—that provide a more successful inductive bias (Neyshabur et al., 2014). For instance, it has been shown that linear models optimized with SGD converge to solutions with small norm, without any explicit regularization (Zhang et al., 2017). In the remainder of the paper, we present a set of experiments that shed more light on the advantages of data augmentation over weight decay and dropout.

## 4 METHODS

This section describes the experimental setup for systematically analyzing the role of data augmentation in deep neural networks compared to weight decay and dropout and builds upon the methods used in preliminary studies (Hernández-García & König, 2018a;b; Zhang et al., 2017).

### 4.1 NETWORK ARCHITECTURES

We perform our experiments on three distinct, popular architectures that have achieved successful results in object recognition tasks: the all convolutional network, All-CNN (Springenberg et al.,

---

[2]Note that the opposite view, that is domain-specific data augmentation as explicit regularization, does not apply. Appendix E extends the discussion on the regularization taxonomy, including the difference between data augmentation and data-dependent regularization

2014); the wide residual network, WRN (Zagoruyko & Komodakis, 2016); and the densely connected network, DenseNet (Huang et al., 2017). Importantly, we keep the same training hyper-parameters (learning rate, training epochs, batch size, optimizer, etc.) as in the original papers in the cases they are reported. Below we present the main features of each network and more details can be found in the supplementary material.

- **All-CNN**: it consists only of convolutional layers with ReLU activation (Glorot et al., 2011), it is relatively shallow and has few parameters. For ImageNet, the network has 16 layers and 9.4 million parameters; for CIFAR, it has 12 layers and 1.3 million parameters. In our experiments to compare the adaptability of data augmentation and explicit regularization to changes in the architecture, we also test a *shallower* version, with 9 layers and 374,000 parameters, and a *deeper* version, with 15 layers and 2.4 million parameters.
- **WRN**: a residual network, ResNet (He et al., 2016), that achieves better performance with fewer layers, but more units per layer. Here, we choose for our experiments the WRN-28-10 version (28 layers and about 36.5 M parameters), which is reported to achieve the best results on CIFAR.
- **DenseNet**: a network architecture arranged in blocks whose layers are connected to all previous layers, allowing for very deep architectures with few parameters. Specifically, for our experiments we use a DenseNet-BC with growth rate $k = 12$ and 16 layers in each block, which has a total of 0.8 million parameters.

### 4.2 DATA

We perform the experiments on the highly benchmarked data sets ImageNet (Russakovsky et al., 2015) ILSVRC 2012, CIFAR-10 and CIFAR-100 (Krizhevsky & Hinton, 2009). We resize the 1.3 M images from ImageNet into $150 \times 200$ pixels, as a compromise between keeping a high resolution and speeding up the training. Both on ImageNet and on CIFAR, the pixel values are in the range $[0, 1]$ and have 32 bits floating precision.

So as to analyze the role of data augmentation, we train every network architecture with two different augmentation schemes as well as with no data augmentation at all:

- *Light* **augmentation**: This scheme is common in the literature, for example (Goodfellow et al., 2013; Springenberg et al., 2014), and performs only horizontal flips and horizontal and vertical translations of 10% of the image size.
- *Heavier* **augmentation**: This scheme performs a larger range of affine transformations such as scaling, rotations and shear mappings, as well as contrast and brightness adjustment. On ImageNet we additionally perform a random crop of $128 \times 128$ pixels. The choice of the allowed transformations is arbitrary and the only criterion was that the objects are still recognizable in general. We deliberately avoid designing a particularly successful scheme. The details of the *heavier* scheme can be consulted in the supplementary material.

### 4.3 TRAIN AND TEST

Every architecture is trained on each data set both with explicit regularization—weight decay and dropout as specified in the original papers—and with no explicit regularization. Furthermore, we train each model with the three data augmentation schemes. The performance of the models is computed on the held out test tests. As in previous works (Krizhevsky et al., 2012; Simonyan & Zisserman, 2014), we average the softmax posteriors over 10 random *light* augmentations, since slightly better results are obtained.

All the experiments are performed on Keras (Chollet et al., 2015) on top of TensorFlow (Abadi et al., 2015) and on a single GPU NVIDIA GeForce GTX 1080 Ti.

## 5 RESULTS

This section presents the most relevant results of the experiments comparing the roles of data augmentation and explicit regularization on convolutional neural networks. First, we present the

experiments with the original architectures in section 5.1. Then, Sections 5.2 and 5.3 show the results of training the models with fewer training examples and with shallower and deeper versions of the All-CNN architecture.

The figures aim at facilitating the comparison between the models trained with and without explicit regularization, as well as between the different levels of data augmentation. The purple bars (top of each pair) correspond to the models trained *without* explicit regularization—weight decay and dropout—and the red bars (bottom) to the models trained *with* it. The different color shades correspond to the three augmentation schemes. The figures show the relative performance of each model with respect to a particular baseline in order to highlight the relevant comparisons. A detailed and complete report of all the results can be found in the supplementary material. The results on CIFAR refer to the top-1 test accuracy while on ImageNet we report the top-5.

## 5.1 An Alternative to Explicit Regularization

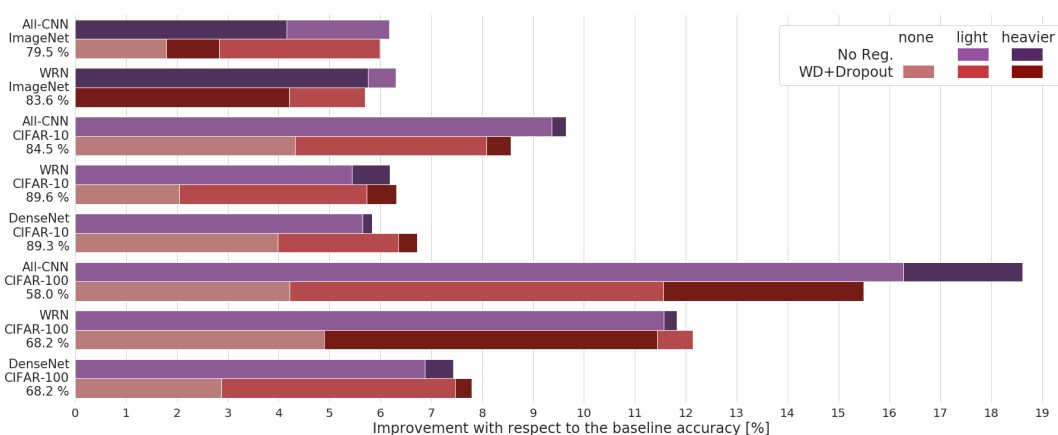

Figure 1: Relative improvement of adding data augmentation and explicit regularization to the baseline models, $(accuracy - baseline)/accuracy * 100$. The baseline accuracy is shown on the left. The results suggest that data augmentation alone (purple bars) can achieve even better performance than the models trained with both weight decay and dropout (red bars).

First, we contrast the regularization effect of data augmentation and weight decay and dropout on the original networks trained with the complete data sets. For that purpose, in Figure 1 we show the relative improvement in test performance achieved by adding each technique or combination of techniques to the baseline model, that is the model trained with neither explicit regularization nor data augmentation (see the left of the bars). Table 1 shows the mean and standard deviation of each combination.[3]

Table 1: Average accuracy improvement over the baseline model of each combination of data augmentation level and presence of weight decay and dropout.

|         | No explicit reg. | Weight decay + dropout |
|---------|------------------|------------------------|
| None    | *baseline*       | 3.02 (1.65)            |
| Light   | 8.46 (3.80)      | 7.88 (2.60)            |
| Heavier | 8.68 (4.69)      | 7.92 (4.03)            |

Several conclusions can be extracted from Figure 1 and Table 1. Most importantly, training with data augmentation alone (top, purple bars) improves the performance in most cases as much as or even more than training with both augmentation and explicit regularization (bottom, red bars), on average

---

[3]The relative performance of WRN on ImageNet trained with weight decay and dropout with respect to the baseline is negative (-6.22 %) and is neither depicted in Figure 1 nor taken into consideration to compute the average improvements in Table 1.

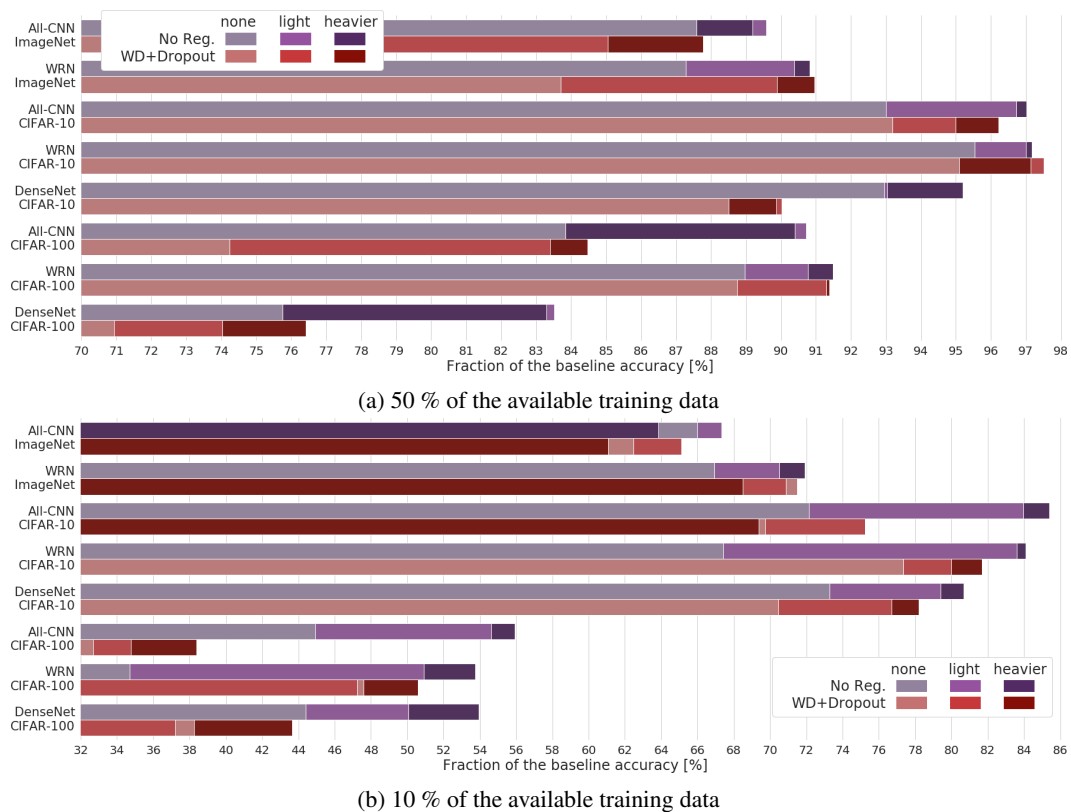

(a) 50 % of the available training data

(b) 10 % of the available training data

Figure 2: Fraction of the baseline performance when the amount of available training data is reduced, $accuracy/baseline * 100$. The models trained wit explicit regularization present a significant drop in performance as compared to the models trained with only data augmentation. The differences become larger as the amount of training data decreases.

8.57 and 7.90 % respectively. This is quite a surprising and remarkable result: note that the studied architectures achieved state-of-the-art results at the moment of their publication and the models included both light augmentation and weight decay and dropout, whose parameters were presumably finely tuned to achieve higher accuracy. The replication of these results corresponds to the middle red bars in Figure 1. We show here that simply removing weight decay and dropout—while even keeping all other hyperparameters intact, see Section 4.1—improves the *formerly state-of-the-art* accuracy in 4 of the 8 studied cases.

Second, it can also be observed that the regularization effect of weight decay and dropout, an average improvement of 3.02 % with respect to the baseline,[1] is much smaller than that of data augmentation. Simply applying light augmentation increases the accuracy in 8.46 % on average.

Finally, note that even though the heavier augmentation scheme was deliberately not designed to optimize the performance, in both CIFAR-10 and CIFAR-100 it improves the test performance with respect to the light augmentation scheme. This is not the case on ImageNet, probably due to the increased complexly of the data set. It can be observed though that the effects are in general more consistent in the models trained without explicit regularization. In sum, it seems that the performance gain achieved by weight decay and dropout can be achieved and often improved by data augmentation alone.

## 5.2 FEWER AVAILABLE TRAINING EXAMPLES

We argue that one of the main drawbacks of explicit regularization techniques is their poor adaptability to changes in the conditions with which the hyperparameters were tuned. To test this hypothesis and contrast it with the adaptability of data augmentation, here we extend the analysis by training the

Table 2: Average fraction of the original accuracy of each corresponding combination of data augmentation level and presence of weight decay and dropout.

|  | 50 % of the training data | | 10 % of the training data | |
|---|---|---|---|---|
|  | No explicit reg. | WD + dropout | No explicit reg. | WD + dropout |
| None | 88.11 (6.27) | 83.20 (9.83) | 58.72 (14.93) | 58.75 (16.92) |
| Light | 91.47 (4.31) | 88.27 (7.39) | 67.55 (14.27) | 60.89 (18.39) |
| Heavier | 91.82 (4.63) | 89.28 (6.63) | 68.69 (13.61) | 61.43 (15.90) |

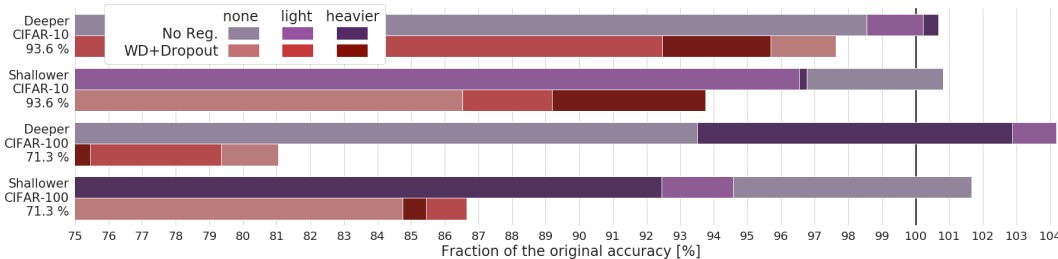

Figure 3: Fraction of the original performance when the depth of the All-CNN architecture is increased or reduced in 3 layers. In the explicitly regularized models, the change of architecture implies a dramatic drop in the performance, while the models trained without explicit regularization present only slight variations with respect to the original architecture.

same networks with fewer examples. The models are trained with the same random subset of data and evaluated in the same test set as the previous experiments. In order to better visualize how well each technique resists the reduction of training data, in Figure 2 we show the fraction of baseline accuracy achieved by each model when trained with 50 % and 10 % of the available data. In this case, the baseline is thus each corresponding model trained with the complete data set. Table 2 summarizes the mean and standard deviation of each combination. An extended report of results, including additional experiments with 80 % and 1 % of the data, is provided in the supplementary material.

One of the main conclusions of this set of experiments is that if no data augmentation is applied, explicit regularization hardly resist the reduction of training data by itself. On average, with 50 % of the available data, these models only achieve 83.20 % of the original accuracy, which, remarkably, is worse than the models trained without any explicit regularization (88.11 %). On 10 % of the data, the average fraction is the same (58.75 and 58.72 %, respectively). This implies that training with explicit regularization is even detrimental for the performance.

When combined with data augmentation, the models trained with explicit regularization (bottom, red bars) also perform worse (88.78 and 61.16 % with 50 and 10 % of the data, respectively), than the models with just data augmentation (top, purple bars, 91.64 and 68.12 % on average). Note that the difference becomes larger as the amount of available data decreases. Importantly, it seems that the combination of explicit regularization and data augmentation is only slightly better than training without data augmentation. We can think of two reasons that could explain this: first, the original regularization hyperparameters seem to adapt poorly to the new conditions. The hyperparameters are specifically tuned for the original setup and one would have to re-tune them to achieve comparable results. Second, since explicit regularization reduces the representational capacity, this might prevent the models from taking advantage of the augmented data.

In contrast, the models trained without explicit regularization more naturally adapt to the reduced availability of data. With 50 % of the data, these models, trained with data augmentation achieve about 91.5 % of the performance with respect to training with the complete data sets. With only 10 % of the data, they achieve nearly 70 % of the baseline performance, on average. This highlights the suitability of data augmentation to serve, to a great extent, as true, useful data (Vinyals et al., 2016).

### 5.3 Shallower and Deeper Architectures

Finally, in this section we test the adaptability of data augmentation and explicit regularization to changes in the depth of the All-CNN architecture (see Section 4.1). We show the fraction of the performance with respect to the original architecture in Figure 3.

A noticeable result from Figure 3 is that all the models trained with weight decay and dropout (bottom, red bars) suffer a dramatic drop in performance when the architecture changes, regardless of whether it becomes deeper or shallower and of the amount of data augmentation. As in the case of reduced training data, this may be explained by the poor adaptability of the regularization hyperparameters, which highly depend on the architecture.

This highly contrasts with the performance of the models trained without explicit regularization (top, purple bars). With a deeper architecture, these models achieve slightly better performance, effectively exploiting the increased capacity. With a shallower architecture, they achieve only slightly worse performance[4]. Thus, these models seem to more naturally adapt to the new architecture and data augmentation becomes beneficial.

It is worth commenting on the particular case of the CIFAR-100 benchmark, where the difference between the models with and without explicit regularization is even more pronounced, in general. It is a common practice in object recognition papers to tune the parameters for CIFAR-10 and then test the performance on CIFAR-100 with the same hyperparameters. Therefore, these are typically less suitable for CIFAR-100. We believe this is the reason why the benefits of data augmentation seem even more pronounced on CIFAR-100 in our experiments.

In sum, these results highlight another crucial advantage of data augmentation: the effectiveness of its hyperparameters, that is the type of image transformations, depend mostly on the type of data, rather than on the particular architecture or amount of available training data, unlike explicit regularization hyperparameters. Therefore, removing explicit regularization and training with data augmentation increases the flexibility of the models.

## 6 Discussion

We have presented a systematic analysis of the role of data augmentation in deep convolutional neural networks for object recognition, focusing on the comparison with popular explicit regularization techniques—weight decay and dropout. In order to facilitate the discussion and the analysis, we first proposed in Section 2 definitions of explicit and implicit regularization, which have been ambiguously used in the literature. Accordingly, we have argued that data augmentation should not be considered an explicit regularizer, such as weight decay and dropout. Then, we provided some theoretical insights in Section 3 that highlight some advantages of data augmentation over explicit regularization. Finally, we have empirically shown that explicit regularization is not only unnecessary (Zhang et al., 2017), but also that its generalization gain can be achieved by data augmentation alone. Moreover, we have demonstrated that, unlike data augmentation, weight decay and dropout exhibit poor adaptability to changes in the architecture and the amount of training data.

Despite the limitations of our empirical study, we have chosen three significantly distinct network architectures and three data sets in order to increase the generality of our conclusions, which should ideally be confirmed by future work on a wider range of models, data sets and even other domains such text or speech. It is important to note, however, that we have taken a conservative approach in our experimentation: all the hyperparameters have been kept as in the original models, which included both weight decay and dropout, as well as light augmentation. This setup is clearly suboptimal for models trained without explicit regularization. Besides, the heavier data augmentation scheme was deliberately not optimized to improve the performance and it was not the scope of this work to propose a specific data augmentation technique. As future work, we plan to propose data augmentation schemes that can more successfully be exploited by any deep model.

---

[4]Note that the shallower models trained with neither explicit regularization nor data augmentation achieve even better accuracy than their counterpart with the original architecture, probably due to the reduction of overfitting provided by the reduced capacity.

The relevance of our findings lies in the fact that explicit regularization is currently the standard tool to enable the generalization of most machine learning methods and is included in most convolutional neural networks. However, we have empirically shown that simply removing the explicit regularizers often improves the performance or only marginally reduces it, if some data augmentation is applied. These results are supported by the theoretical insights provided in in Section 3.

Zhang et al. (2017) suggested that regularization might play a different role in deep learning, not fully explained by statistical learning theory (Vapnik & Chervonenkis, 1971). We have argued instead that the theory still naturally holds in deep learning, as long as one considers the crucial role of implicit regularization: explicit regularization seems to be no longer necessary because its contribution is already provided by the many elements that implicitly and successfully regularize the models: to name a few, stochastic gradient descent, convolutional layers and data augmentation.

### 6.1 RETHINKING DATA AUGMENTATION

Data augmentation is often regarded by authors of machine learning papers as *cheating*, something that should not be used in order to test the potential of a newly proposed architecture (Goodfellow et al., 2013; Graham, 2014; Larsson et al., 2016). In contrast, weight decay and dropout are almost ubiquitous and considered intrinsic elements of the algorithms. In view of the results presented here, we believe that the deep learning community would benefit if we rethink data augmentation and switch roles with explicit regularization: a good model should generalize well without the need for explicit regularization and successful methods should effectively exploit data augmentation.

In this regard it is worth highlighting some of the advantages of data augmentation: Not only does it not reduce the representational capacity of the model, unlike explicit regularization, but also, since the transformations reflect plausible variations of the real objects, it increases the robustness of the model and it can be seen as a data-dependent prior, similarly to unsupervised pre-training (Erhan et al., 2010). Novak et al. (2018) have shown that data augmentation consistently yields models with smaller sensitivity to perturbations. Interestingly, recent work has found that models trained with heavier data augmentation learn representations that are more similar to the inferior temporal (IT) cortex, highlighting the biological plausibility of data augmentation (Hernández-García et al., 2018).

Deep neural networks are especially well suited for data augmentation because they do not rely on pre-computed features and because the large number of parameters allows them to shatter the augmented training set. Moreover, unlike explicit regularization, data augmentation can be performed on the CPU, in parallel to the gradient updates. Finally, an important conclusion from Sections 5.2 and 5.3 is that data augmentation naturally adapts to architectures of different depth and amounts of available training data, whereas explicitly regularized models are highly sensitive to such changes and need specific fine-tuning of their hyperparameters. In sum, data augmentation seems to be a strong alternative to explicit regularization techniques.

Some argue that despite these advantages, data augmentation is a limited approach because it depends on some prior expert knowledge and it cannot be applied to all domains. However, we argue instead that expert knowledge should not be disregarded but exploited. A single data augmentation scheme can be designed for a broad family of data (for example, natural images) and effectively applied to a broad set of tasks (for example, object recognition, segmentation, localization, etc.). Besides, interesting recent works have shown that it is possible to automatically learn the data augmentation strategies (Lemley et al., 2017; Ratner et al., 2017). We hope that these insights encourage more research attention on data augmentation and that future work brings more sophisticated and effective data augmentation techniques, potentially applicable to different data modalities.

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

## A    DETAILS OF NETWORK ARCHITECTURES

This appendix presents the details of the network architectures used in the main experiments: All-CNN, Wide Residual Network (WRN) and DenseNet. All-CNN is a relatively simple, small network with a few number of layers and parameters, WRN is deeper, has residual connections and many more parameters and DenseNet is densely connected and is much deeper, but parameter effective.

### A.1    ALL CONVOLUTIONAL NETWORK

All-CNN consists exclusively of convolutional layers with ReLU activation (Glorot et al., 2011), it is relatively shallow and has few parameters. For ImageNet, the network has 16 layers and 9.4 million parameters; for CIFAR, it has 12 layers and about 1.3 million parameters. In our experiments to compare the adaptability of data augmentation and explicit regularization to changes in the architecture, we also test a *shallower* version, with 9 layers and 374,000 parameters, and a *deeper* version, with 15 layers and 2.4 million parameters. The four architectures can be described as in Table 3, where $K\mathbf{C}D(S)$ is a $D \times D$ convolutional layer with $K$ channels and stride $S$, followed by batch normalization and a ReLU non-linearity. *N.Cl.* is the number of classes and Gl.Avg. refers to global average pooling. The CIFAR network is identical to the All-CNN-C architecture in the original paper, except for the introduction of the batch normalization layers. The ImageNet version also includes batch normalization layers and a stride of 2 instead of 4 in the first layer to compensate for the reduced input size (see below).

Table 3: Specification of the All-CNN architectures.

| | |
|---|---|
| ImageNet | 96**C**11(2)–96**C**1(1)–96**C**3(2)–256**C**5(1) –256**C**1(1)–256**C**3(2)–384**C**3(1) –384**C**1(1)–384**C**3(2)–1024**C**3(1) –1024**C**1(1)–*N.Cl.***C**1(1) –Gl.Avg.–Softmax |
| CIFAR | 2×96**C**3(1)–96**C**3(2)–2×192**C**3(1) –192**C**3(2)–192**C**3(1)–192**C**1(1) –*N.Cl.***C**1(1)–Gl.Avg.–Softmax |
| Shallower | 2×96**C**3(1)–96**C**3(2)–192**C**3(1) –192**C**1(1)–*N.Cl.***C**1(1)–Gl.Avg.–Softmax |
| Deeper | 2×96**C**3(1)–96**C**3(2)–2×192**C**3(1) –192**C**3(2)–2×192**C**3(1)–192**C**3(2) –192**C**3(1)–192**C**1(1) –*N.Cl.***C**1(1)–Gl.Avg.–Softmax |

Importantly, we keep the same training parameters as in the original paper in the cases they are reported. Specifically, the All-CNN networks are trained using stochastic gradient descent, with fixed Nesterov momentum 0.9, learning rate of 0.01 and decay factor of 0.1. The batch size for the experiments on ImageNet is 64 and we train during 25 epochs decaying the learning rate at epochs 10 and 20. On CIFAR, the batch size is 128, we train for 350 epochs and decay the learning rate at epochs 200, 250 and 300. The kernel parameters are initialized according to the Xavier uniform initialization (Glorot & Bengio, 2010).

### A.2    WIDE RESIDUAL NETWORK

WRN is a modification of ResNet (He et al., 2016) that achieves better performance with fewer layers, but more units per layer. Here we choose for our experiments the WRN-28-10 version (28 layers and about 36.5 M parameters), which is reported to achieve the best results on CIFAR. It has the following architecture:

$$16\mathbf{C}3(1)\text{–}4\times160\mathbf{R}\text{–}4\times320\mathbf{R}\text{–}4\times640\mathbf{R}\text{–}BN\text{–}ReLU\text{–}Avg.(8)\text{–}FC\text{–}Softmax$$

where $K\mathbf{R}$ is a residual block with residual function BN–ReLU–$K\mathbf{C}3(1)$–BN–ReLU–$K\mathbf{C}$ 3(1). BN is batch normalization, Avg.(8) is spatial average pooling of size 8 and FC is a fully connected layer. On ImageNet, the stride of the first convolution is 2. The stride of the first convolution within the residual blocks is 1 except in the first block of the series of 4, where it is set to 2 in order to subsample the feature maps.

Similarly, we keep the training parameters of the original paper: we train with SGD, with fixed Nesterov momentum 0.9 and learning rate of 0.1. On ImageNet, the learning rate is decayed by 0.2 at epochs 8 and 15 and we train for a total of 20 epochs with batch size 32. On CIFAR, we train with a batch size of 128 during 200 epochs and decay the learning rate at epochs 60, 120 and 160. The kernel parameters are initialized according to the He normal initialization (He et al., 2015).

### A.3 DENSENET

The main characteristic of DenseNet (Huang et al., 2017) is that the architecture is arranged into blocks whose layers are connected to all the layers below, forming a dense graph of connections, which permits training very deep architectures with fewer parameters than, for instance, ResNet. Here, we use a network with bottleneck compression rate $\theta = 0.5$ (DenseNet-BC), growth rate $k = 12$ and 16 layers in each of the three blocks. The model has nearly 0.8 million parameters. The specific architecture can be descried as follows:

$$2\times k\mathbf{C}3(1)\text{–DB}(16)\text{–TB–DB}(16)\text{–TB–DB}(16)\text{–BN–Gl.Avg.–FC–Softmax}$$

where DB($c$) is a dense block, that is a concatenation of $c$ convolutional blocks. Each convolutional block is of a set of layers whose output is concatenated with the input to form the input of the next convolutional block. A convolutional block with bottleneck structure has the following layers:

$$\text{BN–ReLU–}4\times k\mathbf{C}1(1)\text{–BN–ReLU–}k\mathbf{C}3(1)\text{–Concat.}$$

TB is a transition block, which downsamples the size of the feature maps, formed by the following layers:

$$\text{BN–ReLU–}k\mathbf{C}1(1)\text{–Avg.}(2).$$

Like with All-CNN and WRN, we keep the training hyper-parameters of the original paper. On the CIFAR data sets, we train with SGD, with fixed Nesterov momentum 0.9 and learning rate of 0.1, decayed by 0.1 on epochs 150 and 200 and training for a total of 300 epochs. The batch size is 64 and the are initialized with He initialization.

## B DETAILS OF THE HEAVIER DATA AUGMENTATION SCHEME

In this appendix we present the details of the heavier data augmentation scheme, introduced in Section 3.2:

- Affine transformations:
$$\begin{bmatrix} x' \\ y' \\ 1 \end{bmatrix} = \begin{bmatrix} f_h z_x \cos(\theta) & -z_y \sin(\theta + \phi) & t_x \\ z_x \sin(\theta) & z_y \cos(\theta + \phi) & t_y \\ 0 & 0 & 1 \end{bmatrix} \begin{bmatrix} x \\ y \\ 1 \end{bmatrix}$$
- Contrast adjustment: $x' = \gamma(x - \bar{x}) + \bar{x}$
- Brightness adjustment: $x' = x + \delta$

## C DETAILED AND EXTENDED EXPERIMENTAL RESULTS

This appendix details the results of the main experiments shown in Figures 1, 2 and 3 and provides the results of many other experiments not presented above in order not to clutter the visualization. Some of these results are the top-1 accuracy on ImageNet, the results of the models trained with dropout, but without weight decay; and the results of training with 80 % and 1 % of the data.

Table 4: Description and range of possible values of the parameters used for the heavier augmentation. $B(p)$ denotes a Bernoulli distribution and $\mathcal{U}(a, b)$ a uniform distribution.

| Parameter | Description | Range |
|-----------|-------------|-------|
| $f_h$ | Horiz. flip | $1 - 2B(0.5)$ |
| $t_x$ | Horiz. translation | $\mathcal{U}(-0.1, 0.1)$ |
| $t_y$ | Vert. translation | $\mathcal{U}(-0.1, 0.1)$ |
| $z_x$ | Horiz. scale | $\mathcal{U}(0.85, 1.15)$ |
| $z_y$ | Vert. scale | $\mathcal{U}(0.85, 1.15)$ |
| $\theta$ | Rotation angle | $\mathcal{U}(-22.5°, 22.5°)$ |
| $\phi$ | Shear angle | $\mathcal{U}(-0.15, 0.15)$ |
| $\gamma$ | Contrast | $\mathcal{U}(0.5, 1.5)$ |
| $\delta$ | Brightness | $\mathcal{U}(-0.25, 0.25)$ |

Additionally, for many experiments we also train a version of the network without batch normalization. These results are provided within brackets in the tables. Note that the original All-CNN results published by Springenberg et al. (2014) did not include batch normalization. In the case of WRN, we remove all batch normalization layers except the top-most one, before the spatial average pooling, since otherwise many models would not converge.

Table 5: Test accuracy of All-CNN and WRN, comparing the performance with and without explicit regularizers and the different augmentation schemes. Results within brackets show the performance of the models without batch normalization

| Network | WD | Dropout | Aug. | CIFAR-10 | CIFAR-100 | Acc. ImageNet |
|---------|-----|---------|------|----------|-----------|---------------|
| | yes | yes | no | 90.04 (88.35) | 66.50 (60.54) | 58.09 |
| | yes | yes | light | 93.26 (91.97) | 70.85 (65.57) | 63.35 |
| | yes | yes | heavier | 93.08 (92.44) | 70.59 (68.62) | 60.15 |
| | no | yes | no | 77.99 (87.59) | 52.39 (60.96) | — |
| All-CNN | no | yes | light | 77.20 (92.01) | 69.71 (68.01) | — |
| | no | yes | heavier | 88.29 (92.18) | 70.56 (68.40) | — |
| | no | no | no | 84.53 (71.98) | 57.99 (39.03) | 56.53 |
| | no | no | light | 93.26 (90.10) | 69.26 (63.00) | 63.79 |
| | no | no | heavier | 93.55 (91.48) | 71.25 (71.46) | 61.37 |
| | yes | yes | no | 91.44 (89.30) | 71.67 (67.42) | 54.67 |
| | yes | yes | light | 95.01 (93.48) | 77.58 (74.23) | 68.84 |
| | yes | yes | heavier | 95.60 (94.38) | 76.96 (74.79) | 66.82 |
| | no | yes | no | 91.47 (89.38) | 71.31 (66.85) | — |
| WRN | no | yes | light | 94.76 (93.52) | 77.42 (74.62) | — |
| | no | yes | heavier | 95.58 (94.52) | 77.47 (73.96) | — |
| | no | no | no | 89.56 (85.45) | 68.16 (59.90) | 61.29 |
| | no | no | light | 94.71 (93.69) | 77.08 (75.27) | 69.80 |
| | no | no | heavier | 95.47 (94.95) | 77.30 (75.69) | 69.30 |

An important observation from Table 5 is that the interaction of weight decay and dropout is not always consistent, since in some cases better results can be obtained with both explicit regularizers active and in other cases, only dropout achieves better generalization. In contrast, the effect of data augmentation seems to be consistent: just some light augmentation achieves much better results than training only with the original data set and performing heavier augmentation almost always further improves the test accuracy, without the need for explicit regularization.

Not surprisingly, batch normalization also contributes to improve the generalization of All-CNN and it seems to combine well with data augmentation. On the contrary, when combined with explicit regularization the results are interestingly not consistent in the case of All-CNN: it seems to improve the generalization of the model trained with both weight decay and dropout, but it drastically reduces the performance with only dropout, in the case of CIFAR-10 and CIFAR-100 without augmentation.

A probable explanation is, again, that the regularization hyperparameters would need to be readjusted with a change of the architecture.

Furthermore, it seems that the gap between the performance of the models trained with and without batch normalization is smaller when they are trained without explicit regularization and when they include heavier data augmentation. This can be observed in Table 5, as well as in Table 6, which contains the results of the models trained with fewer examples. It is important to note as well the benefits of batch normalization for obtaining better results when training with fewer examples. However, it is surprising that there is only a small drop in the performance of WRN—95.47 % to 94.95 % without regularization— from removing the batch normalization layers of the residual blocks, given that they were identified as key components of ResNet (He et al., 2016; Zagoruyko & Komodakis, 2016).

Table 6: Test accuracy of All-CNN and WRN when training with only 80 %, 50 %, 10 % and 1 % of the available examples. Results within brackets correspond to the models without batch normalization

| Pct. Data | Expl. Reg. | Aug. scheme | Test CIFAR-10 | | Test CIFAR-100 | |
|---|---|---|---|---|---|---|
| | | | All-CNN | WRN | All-CNN | WRN |
| 80 % | yes | no | 89.41 (86.61) | 90.27 | 63.93 (52.51) | 70.41 |
| | yes | light | 92.20 (91.25) | 94.07 | 67.63 (63.24) | 75.66 |
| | yes | heavier | 92.83 (91.42) | 94.57 | 68.01 (65.89) | 75.51 |
| | no | no | 83.04 (75.00) | 88.98 | 55.78 (35.95) | 66.10 |
| | no | light | 92.25 (88.75) | 93.97 | 69.05 (56.81) | 75.07 |
| | no | heavier | 92.80 (90.55) | 94.84 | 69.40 (63.57) | 75.38 |
| 50 % | yes | no | 85.88 (82.33) | 86.96 | 58.24 (44.94) | 63.60 |
| | yes | light | 90.30 (87.37) | 92.65 | 61.03 (54.68) | 70.83 |
| | yes | heavier | 90.09 (88.94) | 92.86 | 63.25 (57.91) | 70.33 |
| | no | no | 78.61 (69.46) | 85.56 | 48.62 (31.81) | 60.64 |
| | no | light | 90.21 (84.38) | 91.87 | 62.83 (47.84) | 69.97 |
| | no | heavier | 90.76 (87.44) | 92.77 | 64.41 (55.27) | 70.72 |
| 10 % | yes | no | 67.19 (61.61) | 70.73 | 33.77 (19.79) | 34.11 |
| | yes | light | 76.03 (69.18) | 76.00 | 38.51 (22.79) | 36.65 |
| | yes | heavier | 78.69 (64.14) | 78.10 | 38.34 (26.29) | 38.93 |
| | no | no | 60.97 (41.07) | 60.39 | 26.05 (17.55) | 23.65 |
| | no | light | 78.29 (67.65) | 79.19 | 37.84 (24.34) | 39.24 |
| | no | heavier | 79.87 (70.64) | 80.29 | 39.85 (26.31) | 41.44 |
| 1 % | yes | no | 27.53 (29.90) | 33.45 | 9.16 (3.60) | 7.47 |
| | yes | light | 37.18 (26.85) | 34.13 | 9.64 (3.65) | 7.50 |
| | yes | heavier | 42.73 (26.87) | 41.02 | 9.14 (2.52) | 8.37 |
| | no | no | 38.89 (35.68) | 38.63 | 9.50 (5.51) | 9.47 |
| | no | light | 44.35 (29.29) | 43.84 | 9.87 (5.36) | 9.91 |
| | no | heavier | 47.60 (33.72) | 47.14 | 11.45 (3.57) | 11.03 |

The results in Table 6 clearly support the conclusion presented in Section 4.2: data augmentation alone better resists the lack of training data compared to explicit regularizers. Already with 80% and 50% of the data better results are obtained in some cases, but the differences become much bigger when training with only 10% and 1% of the available data. It seems that explicit regularization prevents the model from both fitting the data and generalizing well, whereas data augmentation provides useful transformed examples. Interestingly, with only 1% of the data, even without data augmentation the models without explicit regularization perform better.

The same effect can be observed in Table 7, where both the shallower and deeper versions of All-CNN perform much worse when trained with explicit regularization, even when trained without data augmentation. This is another piece of evidence that explicit regularization needs to be used very carefully, it requires a proper tuning of the hyperparameters and is not always beneficial.

Table 7: Test accuracy of the shallower and deeper versions of All-CNN on CIFAR-10 and CIFAR-100. Results in parentheses show the difference with respect to the original model.

| Expl. Reg. | Aug. | Test CIFAR-10 | | Test CIFAR-100 | |
|:---:|:---:|:---:|:---:|:---:|:---:|
| | | Shallower | Deeper | Shallower | Deeper |
| yes | no | 76.45 (-13.59) | 86.26 (-3.78) | 51.31 (-9.23) | 49.06 (-11.48) |
| yes | light | 82.02 (-11.24) | 85.04 (-8.22) | 56.81 (-8.76) | 52.03 (-13.54) |
| yes | heavier | 86.66 (-6.42) | 88.46 (-4.62) | 58.64 (-9.98) | 51.78 (-16.84) |
| no | no | 85.22 (+0.69) | 83.30 (-1.23) | 58.95 (+0.96) | 54.22 (-3.77) |
| no | light | 90.02 (-3.24) | 93.46 (+0.20) | 65.51 (-3.75) | 72.16 (+2.90) |
| no | heavier | 90.34 (-3.21) | 94.19 (+0.64) | 65.87 (-5.38) | 73.30 (+2.35) |

## D    NORM OF THE WEIGHT MATRIX

In this appendix we provide the computations of the Frobenius norm of the weight matrices of the models trained with different levels of explicit regularization and data augmentation, as a rough estimation of the complexity of the learned models. Table 8 shows the Frobenius norm of the weight matrices of the models trained with different levels of explicit regularization and data augmentation. The clearest conclusion is that heavier data augmentation seems to yield solutions with larger norm. This is always true except in some All-CNN models trained without batch normalization. Another observation is that, as expected, weight decay constrains the norm of the learned function. Besides, the models trained without batch normalization exhibit smaller differences between different levels of regularization and augmentation and, in the case of All-CNN, less consistency.

Table 8: Frobenius norm of the weight matrices learned by the networks All-CNN and WRN on CIFAR-10 and CIFAR-100, trained with and without explicit regularizers and the different augmentation schemes. Norms within brackets correspond to the models without batch normalization

| WD | Dropout | Aug. | Norm CIFAR-10 | | Norm CIFAR-100 | |
|:---:|:---:|:---:|:---:|:---:|:---:|:---:|
| | | | All-CNN | WRN | All-CNN | WRN |
| yes | yes | no | 48.7 (64.9) | 101.4 (122.6) | 76.5 (97.9) | 134.8 (126.5) |
| yes | yes | light | 52.7 (63.2) | 106.1 (123.9) | 77.6 (86.8) | 140.8 (129.3) |
| yes | yes | heavier | 57.6 (62.8) | 119.3 (125.3) | 78.1 (83.1) | 164.2 (132.5) |
| no | yes | no | 52.4 (70.5) | 153.3 (122.5) | 79.7 (103.3) | 185.1 (126.5) |
| no | yes | light | 57.0 (67.9) | 160.6 (123.9) | 83.6 (93.0) | 199.0 (129.4) |
| no | yes | heavier | 62.8 (67.5) | 175.1 (125.2) | 84.0 (88.0) | 225.4 (132.5) |
| no | no | no | 37.3 (63.7) | 139.0 (120.4) | 47.6 (102.7) | 157.9 (122.0) |
| no | no | light | 47.0 (69.5) | 153.6 (123.2) | 80.0 (108.9) | 187.0 (127.2) |
| no | no | heavier | 62.0 (71.7) | 170.4 (125.4) | 91.7 (91.7) | 217.6 (132.9) |

One of the relevant results presented in this paper is the poor performance of the regularized models on the shallower and deeper versions of All-CNN, compared to the models without explicit regularization (see Table 7). One hypothesis is that the *amount* of regularization is not properly adjusted through the hyperparameters. This could be reflected in the norm of the learned weights, shown in Table 9. However, the norm alone does not seem to fully explain the large performance differences between the different models. Finding the exact reasons why the regularized models not able to generalize well might require a much thorough analysis and we leave it as future work.

## E    ON THE TAXONOMY OF REGULARIZATION

Although it is out of the scope of this paper to elaborated on the taxonomy of regularization techniques for deep neural networks, an important contribution of this work is providing definitions of explicit and implicit regularization, which have been used ambiguously in the literature before. It is therefore worth mentioning here some of the previous works that have used these terms and to point to literature that has specifically elaborated on the regularization taxonomy or proposed other related terms.

Table 9: Frobenius norm of the weight matrices learned by the shallower and deeper versions of the All-CNN network on CIFAR-10 and CIFAR-100.

| Explicit Reg. | Aug. scheme | Norm CIFAR-10 | | Norm CIFAR-100 | |
|---|---|---|---|---|---|
| | | Shallower | Deeper | Shallower | Deeper |
| yes | no | 47.9 | 62.3 | 68.9 | 92.1 |
| yes | light | 49.7 | 66.5 | 67.1 | 95.7 |
| yes | heavier | 51.9 | 71.5 | 66.2 | 96.9 |
| no | no | 34.8 | 45.4 | 64.7 | 53.4 |
| no | light | 45.6 | 57.3 | 68.8 | 77.3 |
| no | heavier | 53.1 | 70.7 | 68.3 | 97.5 |

Neyshabur et al. (2014) observed that the size of neural networks could not explain and control by itself the effective capacity of neural networks and proposed that other elements should implicitly regularize the models. However, no definitions or clear distinction between explicit and implicit regularization was provided. Later, Zhang et al. (2017) compared different regularization techniques and mentioned the role of implicit regularization, but did not provide definitions either, and, importantly, they considered data augmentation an explicit form of regularization. We have argued against that view throughout this paper, especially in Sections 2 and 6.1.

An extensive review of the taxonomy of regularization techniques was carried out by Kukačka et al. (2017). Although no distinction is made between explicit and implicit regularization, they define the class *regularization via optimization*, which is somehow related to implicit regularization. However, regularization via optimization is more specific than our definition and data augmentation, among others, would not fall into that category.

Recently, Guo et al. (2018) provided a distinction between *data-independent* and *data-dependent* regularization. They define data-independent regularization as those techniques that impose certain constraint on the hypothesis set, thus constraining the optimization problem. Examples are weight decay and dropout. We believe this is closely related to our definition of explicit regularization. Then, they define data-dependent regularization as those techniques that make assumptions on the hypothesis set with respect to the training data, as is the case of data augmentation.

While we acknowledge the usefulness of such taxonomy, we believe the division between data-independent and dependent regularization leaves some ambiguity about other techniques, such as batch-normalization, which neither imposes an explicit constraint on H nor on the training data. The taxonomy of explicit vs. implicit regularization is however complete, since implicit regularization refers to any regularization effect that does not come from explicit (or data-independent) techniques.

Finally, we argue it would be useful to distinguish between domain-specific, perceptually-motivated data augmentation and other kinds of data-dependent regularization. Data augmentation ultimately aims at creating new examples that could be plausible transformations of the real-world objects. In other words, the augmented samples should be no different in nature than the available data. In statistical terms, they should belong to the same underlying probability distribution. In contrast, one can think of data manipulations that would not mimic any plausible transformation of the data, which still can improve generalization and thus fall into the category of data-dependent regularization (and implicit regularization). One example is mixup, which is the subject of study of Guo et al. (2018).

