# OpenReview forum: "Data augmentation instead of explicit regularization"
_ICLR.cc/2020/Conference — Reject_

### Official Review · AnonReviewer1 · 2019-10-14
**Official Blind Review #1**

**Rating:** 1

**Review:**

This paper proposed to adopt data augmentation over explicit regularization (weight decay and dropout), supported by a comparative study of 3 ConvNets (all-conv, wide-resnet and densenet) on three image classification benchmarks (ImageNet, CIFAR-10 and CIFAR-100), as well as variants of experiments such as training with smaller training set.

Most of the observations presented here are largely known to the community, though I think at least a few of them are worth emphasizing:

1. One major downside of explicit regularization the requirement to tune hyper-parameters on each different scenarios. For example, the paper shows that the default hyper-parameter becomes suboptimal when the architecture becomes either deeper or shallower.
2. Given strong enough data augmentation, sometimes turning off weight decay / dropout could give better test performance (i.e. the optimal hyper-parameters are zeros in those cases).

That being said, I think this the paper at the current state does not contain sufficient message to stand as a full paper. I list a few areas that I think could potentially improve the paper if properly addressed:

1. Proper definition of explicit / implicit regularization: one goal this paper is trying to achieve is an unambiguous definition of explicit / implicit regularization. However, the current definition given by the paper is not any less ambiguous than prevision conventions. If I understand correctly, the paper defines explicit regularization as mechanism that is explicitly designed to reduce the model capacity, and implicit one as anything else (that happens to improve generalization). There is still a lot ambiguity here: for example, one technique, say, random left-right flip of the inputs, is considered implicit regularization because it improves CIFAR-10 classification performance. However, applying it to wrong data, e.g. speech recognition, would potentially hurt the test performance. Would this technique sometimes be an implicit regularization and sometimes not? Another question is regarding dropout, which is classified as explicit regularization by the paper. Yet, if dropout is applied not to some intermediate layer, but to the input layer, does it suddenly become more similar to data augmentation (think of, e.g. cutout augmentation), and therefore an *implicit* regularization? I think a if clear, mathematical and formally verifiable definition of different types of regularization, if possible, will definite make the paper stronger.

2. Theoretical analysis: currently the paper shows the definition of the Rademacher complexity and some handwavy discussions. It does not provide any additional insights apart from what the definition says: explicit regularization constrains the capacity and data augmentation increase the number of training examples. The discussion mentioned that the augmented data are non i.i.d. which would be an interesting topic for providing theoretical insights, but it is ruled "out of the scope of the paper". Besides, the definition of the Rademacher complexity captures the capacity of the hypothesis via the mathematical sup operator, and is completely independent of the underlying algorithm used to compute the sup. Therefore, data augmentation, which is part of the training procedure, does not change the Rademacher complexity, unless we completely re-define the hypothesis space to be a complicated notion that captures something like "all the functions achievable by SGD with this augmentation and that hyper-parameters, etc.". But this will turn the "implicit regularization" nature of "data augmentation" into the regime of "explicit regularization", because it is now confining the hypothesis space into a subset (so reducing the capacity).

In summary, data augmentation clearly improves the generalization performance, but to formally characterize it, one needs an alternative approach from the default Rademacher complexity way. It would make the paper much stronger if a viable approach is proposed, and even stronger if the non-i.i.d. nature of augmented data points could be discussed under the proposed framework.

3. Empirical studies: I think the paper could still be strong even if without any theoretical characterization, if it contains strong supporting experiments. However, currently the paper only studies image classification tasks and a few convolutional neural networks. Moreover, even within those domain and tasks, there are many well known techniques, such as the cutout augmentation, that are not studied. It would be of a much useful paper to the community if the paper could provide a comprehensive survey of existing explicit / implicit regularization techniques across multiple tasks and domains (e.g. the paper mentioned vision, speech recognition, NLP, etc.). Ideally, pros and cons of each regularization technique could be discussed and if the message that "data augmentation is better than explicit regularization" could hold across multiple domains and tasks, then this will definitely be delivering a strong message. Even if that is not the case, this paper could still serve as practical guides to practitioners for choosing between different approaches of regularization techniques.

Other potential improvements are:

a. hyper-parameter tuning for each regularization technique. The paper acknowledged that the hyper-parameters are suboptimal when they change settings for explicit regularization. This is demonstrating a good point that explicit regularization is inconvenient. However, it would be better if in parallel with the default hyper-parameter, we could learn what would be the best results if we re-tune the best hyper-parameters. Because sometimes people would like to achieve the best performance in practice with all the resources and hammers they could get their hands on. So, for example, with strong data augmentation, would the (re-tuned) optimal dropout rate actually perform even better? (if true, this is also contradicting with the main message of the paper)

b. to some extent data augmentation also have "hyper-parameters", which control how to augment the data. It would be interesting to see, for example, controlled experiments on the effects on test performance when the "wrong" type or magnitudes of augmentation is applied to the data.

minor things:

* all bars in the plots are red / purplish colors. Maybe more diverse colors could be used to make it easier to distinguish which is which?
* Why is there no DenseNet results on ImageNet? (e.g. Figure 1 missing one row?)
* As a paper that studies augmentation, it would be great to provide full details on all the details (including the hyper parameters on the magnitudes of all perturbations) of the data augmentation used, especially for the "heavier augmentation" variant.

Finally, I'm not sure if this violate the policy of anonymous submission --- the acknowledgements section mentioned explicit names when thanking for feedback.

**Experience Assessment:**

I have published in this field for several years.

**Review Assessment: Checking Correctness Of Derivations And Theory:**

N/A

**Review Assessment: Checking Correctness Of Experiments:**

I assessed the sensibility of the experiments.

**Review Assessment: Thoroughness In Paper Reading:**

I read the paper at least twice and used my best judgement in assessing the paper.

---

> ### Author Response · Authors · 2019-11-12
> **Updated definitions of explicit and implicit regularization (#1)**
>
> We sincerely appreciate the thorough review of our paper and thank the reviewer for challenging some aspects that are surely helping improve the manuscript. We are currently updating the paper by addressing the points made in the review and will soon write an extended response highlighting these and replying all the reviewer's concerns. However, we first addressed the topic of the definitions of explicit and implicit regularization ("1. Proper definition of explicit / implicit regularization"), since it was mentioned by two reviewers and we agree on their concerns. We would highly appreciate if the reviewer could check the corresponding part of the updated manuscript and let us know if it satisfies their concerns or provide further feedback otherwise. Thanks in advance.
>
> We agree that the definitions of explicit and implicit regularization in the submitted version of the paper could be improved. We have taken the reviewers' feedback to provide new, more formal and mathematically grounded definitions, while keeping the essence of the original definitions. We have also included examples of each category for better illustration. Please see Section 2 of the updated manuscript. In sum, we now define explicit regularization techniques as those that reduce the *representational* capacity of a model class H and implicit regularization as the improved generalization effect provided by means other than explicit regularization.
>
> The reviewer mentions two examples to challenge the definition of implicit regularization:
>
> 1) "random left-right flip of the inputs, is considered implicit regularization because it improves CIFAR-10 classification performance. However, applying it to wrong data, e.g. speech recognition, would potentially hurt the test performance"
>
> Two aspects of the definition of implicit regularization are relevant to this example. First, we highlight that while we refer to explicit regularization *techniques*, we refer to implicit regularization *effect*, since it can be provided by elements of very different nature, not only clearly defined techniques. Second, please note that we define implicit regularization as a "reduction of the generalization error". Therefore, we argue that a particular technique should not be labelled an implicit regularizer in general, since it would only provide regularization *if* it indeed improves the generalization. Still, note that also explicit regularization techniques do not always improve generalization. For instance, too much weight decay or dropout will prevent a model from improving the generalization (as we actually show in our paper) or even from converging.
>
> 2) "dropout is classified as explicit regularization by the paper. Yet, if dropout is applied not to some intermediate layer, but to the input layer, does it suddenly become more similar to data augmentation (think of, e.g. cutout augmentation), and therefore an *implicit* regularization?"
>
> This is an interesting point which helped us improve the definitions and also allows us highlight some important aspects. According to the updated definition, if a technique constrains the representational capacity, then it should be considered explicit regularization. This would indeed be the case of the mentioned example. Would it not also be data augmentation? And, if so, would it not be implicit regularization? The reason why this question can be raised is because of the nature of neural networks, where the input data can also be considered a layer of the architecture. The answer to the second question derives directly from the definitions: no. Since it reduces the representational capacity it is explicit regularization. The second question needs further elaboration: We argue it would be useful to distinguish between actual, domain-specific data augmentation and data-dependent regularization (See [1]), the former being a particular case of the latter. Data augmentation should imply the use of the prior knowledge about the data domain (e.g. natural images). Ultimately, data augmentation seeks to create plausible transformations of the objects, not different in nature (underlying probability distribution) from the existing samples in the data set. These transformations in general do not reduce the representational capacity and therefore cannot be considered explicit regularization. We have included a discussion about this in the Appendix E. Nonetheless, we see no contradiction in some very specific data manipulations reducing the representational capacity and being therefore explicit regularization.
>
> We hope the updated definitions resolve the ambiguity and the discussion provided here and in the paper sheds light on the reviewer's concerns about this issue. We look forward to your feedback.
>
> [1] Hongyu Guo, Yongyi Mao, and Richong Zhang. Mixup as locally linear out-of-manifold regularization. arXiv preprint arXiv:1809.02499, 2018.

---

> ### Author Response · Authors · 2019-11-15
> **On the novelty and the theoretical insights**
>
> We address here the points raised by the reviewer regarding novelty and the theoretical insights.
>
> Novelty:
>
> First, we would like to comment on the statement that "most of the observations presented here are largely known to the community". We assume this refers to the findings that data augmentation alone can outperform models trained with both data augmentation and explicit regularization. It is hard to measure whether something is largely known by a community, but we can state that to the best of our knowledge there is no previous, published work that demonstrates presents these results. Further, if it was largely known by the community, it would be highly remarkable that the community keeps nonetheless using explicit regularization, that is dropout and especially weight decay, while the same results or higher can be obtained without them and, especially, because their hyperparameters are extremely sensitive to changes in the architecture or amount of training data, as we show in our results.
>
> "2. Theoretical insights"
>
> The reviewer sees as a weakness that our manuscript does not elaborate further on the effect of data augmentation on the model's effective capacity and generalization. We totally agree that this is an interesting topic. However, the aim of our paper is the systematic empirical evaluation of the interaction between data augmentation and explicit regularization (which are often used all together), testing several network architectures trained on different data sets. We believe that the inclusion of a section with theoretical insights is a strenght, rather than a weakness, since we provide some relevant definitions, establish connections with statistical learning theory that support the empirical findings and hopefully set some grounds for future work in this direction. The reason why deriving a theoretical framework about the effect of data augmentation on the model's effective capacity and generalization is "out of the scope" is that this is a very complex topic that certainly requires more than a section in a 10-page publication. By way of illustration, let us mention two pieces of work that address this topic: Rajput et al., 2019 [1] presented at the last ICML a paper where they analyzed the effect on data augmentation on the classification margin. Specifically, they theoretically derive the number of data points needed to ensure a positive margin through data augmentation. While it is a hihgly interesting and relevant paper, the authors admit that they had to set several restrictions to enable the analysis, for instance in the type of classifiers and, very importantly, the type of data augmentation: they restrict the analysis to additive spherical data augmentation. In our case, we were interested in analyzing actual, commonly used data augmentation, and that is why we opted for an empirical analysis and "just" some theoretical insigts. In the updated manuscript we refer to this and other publications for further theoretical insights about data augmentation (Section 3).
>
> The reviewer also states that "data augmentation, which is part of the training procedure, does not change the Rademacher complexity". We disagree in this point: By definition, the Rademacher complexity is expressed with respect to a set of data samples S = (x1, ..., xn), as we explicitly write in our paper, borrowed from the statistical learning theory. Therefore, the number of examples n plays a crucial role in the Rademacher complexity, as denoted by the averaging n within the sup operator. Hence, since data augmentation changes the training data, then it does affect the Rademacher complexity. The intuition is that the Rademacher complexity expresses the "richness" of a model class with a hypothesis space H, *with respect to* a given set of data S. In other words, given a model class H, the Rademacher complexity changes if the data changes. Further, as denoted by our updated definitions of explicit and implicit regularization, the distinction between representational capacity and effective capacity is very relevant. We believe that when the reviewer mentions "all the functions achievable by SGD with this augmentation and that hyper-parameters, etc.", this means the effective capacity and not the representational capacity (H) of the model, which is not affected by neither SGD, the hyperparameters, data augmentation, etc. Note that in the updated version we have been especially careful to clearly distinguish between representational and effective capacity.
>
> [1] Shashank Rajput, Zhili Feng, Zachary Charles, Po-Ling Loh, and Dimitris Papailiopoulos. Does data augmentation lead to positive margin? ICML, 2019.

---

> ### Author Response · Authors · 2019-11-15
> **On the breadth of the empirical evaluation and minor concerns**
>
> We address here the rest of the points raised by the reviewer.
>
> "3. Empirical studies"
>
> We agree that more experimentation is always desirable, but we humbly wonder how much is enough to draw meaningful conclusions. We believe that our paper does provide a considerable amount of empirical variability: we train three very different network architectures (DenseNet, ResNet and All-CNN) on three different data sets (ImageNet, CIFAR-10 and CIFAR-100). Further we consider three levels of data augmentation, we train the models with 100, 80, 50, 10 and 1 % of the training data, we test different depths on All-CNN, we switch on and off weight decay, dropout and batch-normalization. Overall, we report the performance of more than 300 different model instances and, very importantly, the same conclusions can be derived consistently from the vast majority of the results. We honestly believe that this amount of variability is at least comparable to most machine learning research papers. Although this is irrelevant for the review, it is however fair to mention that this work has been carried out by one person on one single GPU over many months and, while considerably extending the empirical evaluation is not feasible for the present work, we provide code that should allow the community to carry on further analyses. Furthermore, as previously discussed, we also provide some theoretical insights from statistical learning theory that support the empirical findings.
>
> The scope of our paper is not to analyze particular data augmentation techniques, such as cutout, but rather to analyze the effect of data augmentation in general, regardless of the specific transformations. This is why we chose on the one hand a widely adopted scheme (light augmentation) and, on the other hand, a scheme with multiple, heavier transformations.
>
> Experimentation on other data modalities would be of course desirable. However, not only would that require a lot more work and of very different nature. Nonetheless, we also wonder whether such results would fit a 10-page paper. Note that despite squeezing dozens of results in graphical figures (unlike the common practice in machine learning of presenting the results in tables, which are less readable and need more space), we still had to move many of our results to appendices. Besides, we humbly believe that presenting results across different data modalities is neither very common nor a general requirement in machine learning papers. Many influential papers were published (and accepted) containing only results on image object classification. Just by way of illustration, the original dropout and batch norm papers or the best paper award at ICLR 2017, which served as a source of inspiration for the present work, "Understanding deep learning requires rethinking regularization".
>
> Regarding the minor comments:
>
> - Colours of the bars in the figures: the colours are thought to be distinguishable for colour-blind people and on black and white printing. The key aspect is to distinguish between no augmentation, light and heavier augmentation, since the comparison between regularization and no regularization is straightforward by the arrangement in pairs of bars.
>
> - "Why is there no DenseNet results on ImageNet?": due the computational limitations. One DenseNet model on CIFAR-10 took us 1.2 days of training. Each pair of bars contains results of 6 models. Training on DenseNet on ImageNet is unfeasible for us, but we provide the code and it would not be such a limitation for other people in the community.
>
> - The details of the augmentation schemes are provided in Appendix B. These contain all the parameters, range and type of distribution for the sampling. Further details con be consulted in the code provided.
>
> - We have not seen any policy regarding the mention of people in the acknowledgements, but we have anyway removed it from the updated manuscript.

---

### Official Review · AnonReviewer3 · 2019-10-19
**Official Blind Review #3**

**Rating:** 3

**Review:**

The paper questions the conventional wisdom of using explicit regularization methods (e.g., L2, dropout) in training neural networks. The authors compare data augmentation with explicit regularization on several image classification datasets, architectures and amount of data, concluding using data augmentations is enough to reach a on-par performance with using explicit regularization. I do have several concerns about the paper.

1. My most worrying concerns are about the experiments.
(1) The ImageNet experimental setting is not that convincing since it follows a different resolution than the literature. The results obtained (e.g., 17% top-5 error) are too far from state-of-the-art.

(2) The weight decay and dropout are used together, but not separately studied. In fact, in state-of-the-art CIFAR and ImageNet models, dropout are often not used. Though the reason is that they already use data augmentations, so dropout is typically no longer helpful (WRN is an exception). I think L2 alone is more worth studying, since it is probably known that dropout doesn't help upon a conventional augmentation.

(3) The hyperparameters used for WD+dropout in the experiments are "as specified in the original papers". But the original papers assume the conventional data augmentation. If you use new data augmentation schemes (light/heavier), the regularization hyperparameters should be tuned accordingly. I believe if the strengths of weight decay is properly tuned then it should help even with data augmentation, by a noticeable margin.

(4) If we only see WRN and DenseNet on CIFAR datasets (All-CNN is probably outdated and performs poorly, and ImageNet is not convincing as said in (1)), we notice that actually WD+dropout do provides a small increase on the augmentation schemes. This does not support the main claim of the paper.

(5) More experiments on other domains (e.g., NLP) can be used to strengthen the paper, since the title does not specify a modality.

2. "Explicit regularization techniques ... they blindly reduce the effective capacity of the model, introduce sensitive hyper-parameters and require deeper and wider architectures to compensate for the reduced capacity". I cannot agree the regularization schemes just "blindly" reduce the capacity. Take L2 weight decay as example, it does not reduce the theoretical representation power of the network, all it does is to encourage simpler solutions. Also, data augmentation schemes involve a lot of hyper-parameters too, and possibly requires deeper/wider architectures to fully exploit its advantage. In my opinion, data augmentation does not solve the possible inconvenience brought by explicit regularizations.


3.The definitions of explicit and implicit regularization in Section 2 a bit vague. Under this two definitions, I can see dropout actually falls in both categories, despite slightly more similar to the explicit one. On one hand it is specifically restricting the model's capacity by sampling a smaller model in each iteration, and on the other hand it also changes "the learning algorithm" and "characteristics of the network architecture". Similar thing holds for "Stochastic Depth". Also, injecting noise in intermediate activations is very similar to dropout since dropout is actually injecting noise by randomly removing a portion of the activations. However I can see under these two definitions injecting noise is implicit while dropout is implicit. I think it helps to list at least 5 or 6 examples for each category right there.


In summary, the claims are not well supported by the experiments, and I tend to reject the paper.
--------------------------------------------------------------

I appreciate the detailed author response and have some quick comments:

I couldn't agree with the arguments about hyperparameters of DA. They are meaningful but their influence to the network training still needs to be tuned, possibly for different architectures, to best optimize performance.

It is also not justified that "increasing resolution" won't help in this case since your top-5 error is a bit too high compared with recent results. Especially for a strong argument in the title I would expect a more standard setting evaluated.

The DenseNet paper only uses dropout in absence of data augmentation, and in my personal experience if DA is used dropout is not helpful anymore, for ResNet as well. In standard ImageNet augmentation scheme, no one uses dropout in popular models of recent years (ResNet, DenseNet, SENet, etc.) but all use weight decay.

I'm happy with some other parts of the response, e.g., about the title, update of definitions. Some other points seem more like opinions and I might have a different opinion from the authors, e.g., whether weight decay constrains hypothesis space.

**Experience Assessment:**

I have published one or two papers in this area.

**Review Assessment: Checking Correctness Of Derivations And Theory:**

I carefully checked the derivations and theory.

**Review Assessment: Checking Correctness Of Experiments:**

I carefully checked the experiments.

**Review Assessment: Thoroughness In Paper Reading:**

I read the paper thoroughly.

---

> ### Author Response · Authors · 2019-11-12
> **Updated definitions of explicit and implicit regularization (#3)**
>
> We would first like to thank the reviewer for their feedback and we are confident it is helping us improve the manuscript. We are currently updating the paper by addressing the points made in the review and we will soon write an extended response highlighting these and replying all the reviewer's concerns. However, we first addressed the topic of the definitions of explicit and implicit regularization ("3.The definitions of explicit and implicit regularization in Section 2 a bit vague."), since it was mentioned by two reviewers and we agree on their concerns. We would highly appreciate if the reviewer could check the corresponding part of the updated manuscript and let us know if it satisfies their concerns or provide further feedback otherwise. Thanks in advance.
>
> We agree that the definitions of explicit and implicit regularization in the submitted version of the paper were not unambiguous enough. Accordingly, we have updated Section 2 of the paper with new, more formal and mathematically grounded definitions, while keeping the essence of the original definitions. We believe the new definitions resolve the ambiguity in the examples mentioned by the reviewer. As suggested in the review, we have also included examples of each category for better illustration. Please refer to Section 2 of the updated manuscript for the specifics. In sum, we now define explicit regularization techniques as those that reduce the *representational* capacity of a model class H and implicit regularization as the improved generalization effect provided by means other than explicit regularization.
>
> Under this new light, we can now address the previously problematic cases mentioned by the reviewer: both dropout and stochastic depth constrain the representational capacity of the model by "sampling a smaller model in each iteration", as stated in the review. According to the updated definitions, they both are therefore explicit regularization and, consequently, implicit regularization is ruled out by exclusion. The same would apply for the injection of noise in intermediate activations if it reduces the representational capacity. The definitions reserve implicit regularization for the generalization gains provided by elements that do not constrain the representational capacity, that is explicit regularization. As we have now included in the paper, examples are SGD, batch normalization or early stopping, which may affect the effective, but not the representational capacity. Another relevant example in our paper is of course domain-specific data augmentation, which transforms the input data into plausible new examples and does not affect the hypothesis set spanned by the model class.
>
> We hope the updated definitions resolve the ambiguity, we look forward to hearing the reviewer's thoughts in this regard and we remain open to further, slight modifications.

---

> ### Author Response · Authors · 2019-11-15
> **Experiments and hyperparameters of weight decay and dropout**
>
> We address here other points raised by the reviewer.
>
> 1. Experiments
>
> (1) ImageNet experimental setting following a different resolution than literature.
>
> It has been noted by several pieces of work (see [1], for instance) that the results on ImageNet do not improve significantly beyond 128x128 px resolution, which is what we used. It is fair to mention that we have very limited computational resources (1 GPU) and training on ImageNet is very costly in computational terms. Given the amount of experiments in our analysis (every model/dataset requires training 6 models), increasing the resolution would have dramatically increased the training time, without any guarantee of obtaining better results.
>
> (2) Dropout and weight decay. The reviewer affirms that in state-of-the-art models dropout is often not used. We disagree and can provide some examples and citations statistics to support this. Take, for instance, EfficientNet: it uses not only dropout but also stochastic depth. The paper released a few days ago showing state-of-the-art results on ImageNet [1] also uses dropout. The (main) original dropout paper [2] has a total of 15,625 citations according to Google Scholar, more than half of them (8.650, 55.36 %) since 2018 and 4.990 (31.94 %) in the present year. The chances are that many of this citations are papers applying dropout. In particular, the three network architectures studied in this paper achieved state of the art results at their time of publication and all used both weight decay and dropout. We therefore argue that WRN is not an exception, but the norm. In any case, "if it is probably known that dropout doesn't help upon a conventional augmentation", there is not published evidence of this, to the best of our knowledge.
>
> While the core results in our paper, presented in the main body of the paper, show the results of jointly turning off dropout and weight decay, the appendices include results of training with only weight decay. In the updated version, following the reviewer's suggestion, we have included also results of training with weight decay, without dropout. The results are consistent with the rest of the experiments, that is, the performance is better with only data augmentation and no weight decay.
>
> (3) Hyperparameters of weight decay and dropout
>
> Our "light" augmentation scheme is exactly the same as the one used in the three architectures under evaluation (All-CNN, WRN and DenseNet), as well as in many others in the literature.
>
> The reviewer suggests that if the data augmentation scheme is changed, the hyperparameters of weight decay should be tuned accordingly. This touches an interesting point, that we tried to highlight in the paper: while the "hyperparameters" of data augmentation depend only on the type of data (for instance natural images, text, speech), the hyperparameters of explicit regularization techniques are extremely sensitive to changes in many aspects of the learning process. In our paper, we demonstrate the acute effect of changing the network depth and amount of training data in explicitly regularized models. Although we have no evidence that the performance of weight decay would increase by tuning it hyperparameter according to the data augmentation scheme, even if that was the case, it would highlight a remarkable weakness of weight decay, the hypersensitivity of its hyperparameter. Note that the transformations of data augmentation aim to mimic plausible transformations that would occur in real world images. If that affects the weight decay hyperparameter, it will be bad news for weight decay and would actually strengthen our hypothesis: that we should rather not train with explicit regularization and put more emphasis on developing effective data augmentation schemes.
>
> [1] Chrabaszcz, Patryk, Ilya Loshchilov, and Frank Hutter. "A downsampled variant of imagenet as an alternative to the cifar datasets." arXiv preprint arXiv:1707.08819 (2017).
>
> [2] Srivastava, Nitish, Geoffrey Hinton, Alex Krizhevsky, Ilya Sutskever, and Ruslan Salakhutdinov. Dropout: a simple way to prevent neural networks from overfitting. The Journal of Machine Learning Research 15, no. 1, 2014

---

> ### Author Response · Authors · 2019-11-15
> **Explicit regularization and model capacity**
>
> We address here the comments regarding regularization and model capacity
>
> The reviewer states that weight decay "does not reduce the theoretical representation power of the network". We disagree on this point and argue that reducing the representational power of the network is precisely what weight decay does. If a model that spans a hypothesis set H0 is trained with weight decay, the hypothesis set of the regularized model H1 is necessarily a strict subset of the original one.
>
> The reviewer also states that "all it does is to encourage simpler solutions". We argue that this is actually what reducing the representational power means. Further, we question what "simpler" actually means in a network with tens of parameters, as WRN. This connects with the next point.
>
> By "blindly", we meant that weight decay and dropout just *constrain* the representational capacity to avoid overfitting, as opposed to data augmentation, which performs meaningful, plausible transformations by using prior knowledge about the data domain. Dropout randomly turns off neurons and weight decay sets a penalty on the weight norms. That is arguably more "blind" that creating additional data. In any case, since the term "blindly" is subjective, we have removed from the abstract and rather stick to the formal, mathematical definitions provided in the body of the paper.o
>
> The reviewer notes that data augmentation also involves a lot of hyperparameters, it "possibly requires deeper/wider architectures to fully exploit its advantage" and, in their opinion, "it does not solve the possible inconvenience brought by explicit regularization". While it is true that data augmentation involves a lot of parameters, crucial differences should be taken into account: 1) The hyperparameters of data augmentation reflect meaningul, plausible changes in the real world objects (reasonable rotation, translation, contrast change, etc.) 2) The hyperparameters depend only on the data domain (not on the architecture, optimization algorithm hyperparameters, etc.) As a matter of fact, in our paper, the ~100 models trained with each data augmentation scheme use the same data augmentation "hyperparameters". As opposed to explicit regularization where each tuple architecture & data set has a very different set of hyperparameters. 3) If data augmentation requires deeper/wider architectures, then it means that real-world data would require deeper/wider architectures. Needles to recall that a model should perform well not only on the test set of a benchmark data set, but also on real data, which includes object transformations as the ones used in data augmentation.
>
> We believe that these important differences do address many of the inconvenient aspects of explicit regularization. Data augmentation is not the ultimate solutions for training deep nets for object recognition, but in view of our results and conclusions, it is a very simple technique that offers a wide range of advantages over commonly used explicit regularization techniques.

---

> ### Author Response · Authors · 2019-11-15
> **Breadth of the empirical evaluation and significance in context**
>
> We address here the rest of the points raised by the reviewer.
>
> (4) Architectures and data sets
>
> While All-CNN is older than the other architectures, we believe we should not disregard the results obtained by training it, since it is a purely convolutional neural network (therefore it has a number of advantages and many modern applications use purely convolutional architectures) and it achieves results that are not too far from more modern architectures in a fraction of their complexity. By way of illustration, with our GPU, All-CNN can be trained on CIFAR-10 in about 2 hours, while WRN requires 18 hours and Densenet 1.2 days.
>
> We disagree that the results on Imagenet are not convincing or relevant, as discussed above.
>
> While it is true that WD+dropout provides a small increase on the original architectures, it is important to take into account a few aspects: 1) the performance difference is very small 2) the hyperparameters (learning rate, batch size, etc.) used in the models trained without explicit regularization are the same as in the original papers, trained with explicit regularization and probably highly tuned to push state of the art results. As a small test of the suboptimality of the hyperparameters for the models trained without explicit regularization, we varied the learning rate in a few cases and showed that the performance indeed increased over the baseline models. We decided to keep the original hyperparameters in order to show that our hypothesis holds even in suboptimal scenarios and it was out of the scope of our paper to explore the optimal set of hyperparameters, since that would be of limited scientific insight and computationally costly. 3) Note that the slight performance superiority of WD+Dropout dissapears in the cases where their hyperparameters were not finely tuned (CIFAR-100, depth change and different amount of training data).
>
> (5) Experiments on other data domains.
>
> Experimentation on other data modalities would be of course desirable. However, not only would that require a lot more work and of very different nature, but we also wonder whether the results would fit a 10-page paper. Note that despite squeezing dozens of results in graphical figures (unlike the common practice in machine learning of presenting the results in tables, which are less readable and need more space), we still had to move many of our results to appendices.
>
> Besides, we humbly believe that presenting results across different data modalities is neither very common nor a general requirement in machine learning papers. Many influential papers were published and accepted containing only results on image object classification. Just by way of illustration, the original dropout and batch norm papers or the best paper award at ICLR 2017, which served as a source of inspiration for the present work, "Understanding deep learning requires rethinking regularization".
>
> Regarding the tile not specifying the data domain we study in the paper, we also argue that this is common in machine learning papers. Some examples of widely known, influential papers which only include experiments on image object classification data sets follow:
>
> - Intriguing properties of deep networks
> - Understanding deep learning requires rethinking regularization
> - The lottery ticket hypothesis
> - On the information bottleneck theory of deep learning
> - ...
>
> Nonetheless, we have made explicit in the abstract that our experiments are on image object classification data sets.

---

### Official Review · AnonReviewer2 · 2019-10-23
**Official Blind Review #2**

**Rating:** 3

**Review:**

This paper demonstrates that for regularization, data augmentation usually works better than explicit regularization methods such as weight decay and dropout. The experiments are detailed and also includes theory explanation of why data augmentation works.

Data augmentation (or increasing the size of training data) and explicit regularization are standard methods to overcome overfitting. From my understanding, they are two different methods to tune the model quality, and they can combined to further improve the model performance, as discussed in [1]. So this paper is not well motivated.

[1] DeVries, Terrance, and Graham W. Taylor. "Improved regularization of convolutional neural networks with cutout." arXiv preprint arXiv:1708.04552 (2017).


**Experience Assessment:**

I have read many papers in this area.

**Review Assessment: Checking Correctness Of Derivations And Theory:**

I assessed the sensibility of the derivations and theory.

**Review Assessment: Checking Correctness Of Experiments:**

I assessed the sensibility of the experiments.

**Review Assessment: Thoroughness In Paper Reading:**

I read the paper thoroughly.

---

> ### Author Response · Authors · 2019-11-14
> **On the correct interpretation of the results and the contribution**
>
> We first thank the reviewer for the assessment of our manuscript. We appreciate that some of its strengths have been identified.
>
> In order to frame our response, we would like to recall that the main contribution of our paper is showing that data augmentation alone can achieve the same performance or higher than models trained with both data augmentation and explicit regularization (weight decay and dropout).
>
> The single argument of the reviewer for rejecting the paper is that, according to one paper [1], one particular augmentation technique (cutout) could be combined with explicit regularization to improve the model performance. The reviewer concludes that because of that fact our paper is not well motivated. Arguably, this does not contradict our findings. As we make clear in the paper and can be logically derived, our hypothesis is not that *any* image transformation by itself is enough to outperform the regularization gains provided by weight decay and dropout. Rather, we propose that if *enough* data augmentation is applied, the same performance or higher can be achieved. As a measure of "enough", we take the most common data augmentation scheme in the literature, that is horizontal mirroring and translations of 10 % of the image size. We then demonstrate that turning of the explicit regularization (weight decay and dropout), in a few worst cases only very slightly degrades the performance and in most cases the performance gets higher. This is remarkable since we tested three different architectures which obtained state of the art results at the moment of publication. Note that the authors of those paper could have just removed the explicit regularization from their training procedure and obtain even higher results.
>
> As further motivation for our work, the paper includes a theoretical discussion (Section 3) with insights from statistical learning theory which motivate the hypothesis and support the empirical results.
>
> We hope this clarifies and highlights the contribution of our paper. A similar, yet more elaborated discussion is included in the paper, both in the introduction and the discussion at the end of the paper. Furthermore, we include a discussion about the differences between explicit and implicit regularization in Section 2 and provide definitions which have been improved in the updated version.
>
> [1] DeVries, Terrance, and Graham W. Taylor. "Improved regularization of convolutional neural networks with cutout." arXiv preprint arXiv:1708.04552 (2017).

---

### Public Comment · ~Micah_Goldblum1 · 2019-11-08
**An Interesting Connection**

Hi Authors,
Thank you for your interesting paper.  I wanted to bring to your attention that your insights into explicit regularization is related to our paper which shows that an alternative to weight decay, which stabilizes effective learning rate, can improve performance, especially for networks with batch normalization.[1]  Please consider mentioning the relationship with our work in your next version.

[1] https://arxiv.org/abs/1910.00359

---

> ### Author Response · Authors · 2019-11-12
> **Hi**
>
> Thanks for pointing to your paper. It looks actually related to our work. I'll read it soon and will consider discussing the connections if we find it indeed relevant.

---

### Decision · Program_Chairs · 2019-12-19

**Decision:**

Reject

**Comment:**

The paper explores the setting of *just* using data augmentation without an additional regularization term included.  The submission claims that comparatively good performance can be achieved with data augmentation alone.  The reviewers unanimously felt that the submission was not suitable for publication at ICLR.  The reasons included skepticism that augmentation without regularization is a useful setting to explore, as well as concerns about the experiments used to support the conclusions in the paper.  In particular, there were concerns that the experiments do not match best practice and that the error rates were too high.  Finally, there were concerns about the clarity of definitions of "implicit" and "explicit" regularization.